# Metabolomics and Age-Related Macular Degeneration

**DOI:** 10.3390/metabo9010004

**Published:** 2018-12-27

**Authors:** Connor N. Brown, Brian D. Green, Richard B. Thompson, Anneke I. den Hollander, Imre Lengyel

**Affiliations:** 1Wellcome-Wolfson Institute for Experimental Medicine (WWIEM), Queen’s University Belfast, Belfast BT9 7BL, UK; cbrown88@qub.ac.uk; 2Institute for Global Food Security (IGFS), Queen’s University Belfast, Belfast BT9 6AG, UK; b.green@qub.ac.uk; 3Department of Biochemistry and Molecular Biology, School of Medicine, University of Maryland, Baltimore, MD 21201, USA; rthompson@som.umaryland.edu; 4Department of Ophthalmology, Radboud University Nijmegen Medical Centre, Nijmegen 6525 EX, The Netherlands; a.denhollander@antrg.umcn.nl

**Keywords:** age-related macular degeneration, metabolomics, metabolism, biomarkers, drusen, retinal pigment epithelium

## Abstract

Age-related macular degeneration (AMD) leads to irreversible visual loss, therefore, early intervention is desirable, but due to its multifactorial nature, diagnosis of early disease might be challenging. Identification of early markers for disease development and progression is key for disease diagnosis. Suitable biomarkers can potentially provide opportunities for clinical intervention at a stage of the disease when irreversible changes are yet to take place. One of the most metabolically active tissues in the human body is the retina, making the use of hypothesis-free techniques, like metabolomics, to measure molecular changes in AMD appealing. Indeed, there is increasing evidence that metabolic dysfunction has an important role in the development and progression of AMD. Therefore, metabolomics appears to be an appropriate platform to investigate disease-associated biomarkers. In this review, we explored what is known about metabolic changes in the retina, in conjunction with the emerging literature in AMD metabolomics research. Methods for metabolic biomarker identification in the eye have also been discussed, including the use of tears, vitreous, and aqueous humor, as well as imaging methods, like fluorescence lifetime imaging, that could be translated into a clinical diagnostic tool with molecular level resolution.

## 1. Introduction

Age-related macular degeneration (AMD) accounts for 8.7% of the world’s total blindness and is the leading cause of irreversible visual impairment in the Western world of people aged 65 and older [1,2]. The total number of individuals with this condition is expected to rise to 196 million by 2020 and 288 million by 2040 [2]. Choroidal neovascularization (CNV) is the most aggressive form of advanced AMD and results in the rapid loss of central vision. The other form of advanced AMD, geographic atrophy (GA), is characterized by the progressive loss of central vision due to the death of retinal pigment epithelium (RPE) and photoreceptor cells. At early stages of AMD, accumulation of intracellular lipofuscin in the RPE and the build-up of extracellular deposits under the RPE occurs [3]. Treatment strategies are available for CNV, but not for GA. Antibodies against vascular endothelial growth factor (VEGF) can halt progression of CNV, but the effect is usually not permanent and the disease usually progresses to macular atrophy after the anti-VEGF treatment [4,5].

Age is a predominant risk factor contributing to the development of any AMD, with the prevalence reaching nearly 30% over the age of 85 years in a European population [6]. Environmental factors, such as diet (fat intake and antioxidants) and lifestyle, particularly smoking [7,8], also contribute to the risk of developing the disease. There are several genetic variants that are associated with an increased prevalence of AMD [9]. The two most significant of these are polymorphisms in the *CFH* and *ARMS2* (age-related maculopathy susceptibility 2) genes. The *CFH* gene encodes for complement factor H, a glycoprotein which has an integral role in the regulation of the alternative complement pathway [10]. The *ARMS2* gene [11,12] encodes the ARMS2 protein, which helps to initiate complement activation from the surface of retinal monocytes and microglia by binding to the surface of apoptotic and necrotic cells [13]. More than 50 genetic variants at 34 loci have been associated with AMD development [9]. A large proportion of these genetic variants are located in or near genes of the complement system, lipid metabolism, and extracellular remodeling [14]. These genetic associations, along with risk factors related to diet, serum cholesterol, and triglyceride levels for late-stage AMD [15], highlight the potential importance of studying metabolomic changes to identify systemic molecular biomarkers associated with these risks. In addition, the cellular interactions and exchange of metabolites between the retina, RPE, and choroid complex also provide an important basis of studying local metabolomic alterations in AMD. Considering that the metabolome is closer to the molecular phenotype than either the genome, the transcriptome, or the proteome [16], the study of metabolites could lead to better prediction of the resulting phenotype than other -omics approaches. With AMD progression increasingly associated with metabolic dysfunction, there is the potential for systemic and local metabolic biomarkers to provide opportunities to detect and follow the progression of the disease at an early stage. Systemic biomarkers and the role of lipid metabolism in AMD has been comprehensively highlighted [17,18], whilst general reviews have briefly covered the metabolomics of AMD, alongside other ocular diseases, and the limitations associated with such studies [19,20,21].

The aim of this review is to specifically highlight metabolic processes occurring within the retina, including those related to the development of AMD. It will also provide an in-depth overview of metabolomics studies conducted in AMD. The possible biofluids and metabolomics methods will also be discussed, as well as the potential utility of metabolomics for discovering biomarkers and identifying new therapeutic approaches for AMD and related diseases.

## 2. Metabolic Processes in the Posterior Eye

### 2.1. Energy Sources in the Retina

The complex cellular interactions in the retina give rise to a unique metabolic environment, which is influenced by a range of external factors, including the differential blood supply to the layers of the retina and the detection of light or darkness by the photoreceptors [22]. Due to the number of cell types, there are various metabolic processes which occur throughout the vertebrate retina [23]. Glucose is the primary fuel source for the photoreceptors in the retina, supplied by the choriocapillaris through the Bruch’s membrane (BrM) and the RPE. The metabolic environment of the retina is diverse and increases in complexity with the laminated morphology present between the cells. For instance, the glucose concentration decreases from the RPE surface to the retina surface in vitro [24,25]. Aerobic glycolysis is the primary form of energy metabolism in the retina [26], where glucose is converted to lactate at a comparable rate to cancer cells [27], even when there is plenty of oxygen. In avascular retinas [28], an alternative energy source to adenosine triphosphate (ATP) is required as ATP is exposed to highly active ATP degrading ion pumps between the centrally located photoreceptor mitochondria and the photoreceptor synaptic terminal [29,30]. The phosphocreatine shuttle is instead used [30] and although this is not essential in vascularized mouse retinas, an isoform of the creatine kinase is still localized at the photoreceptor synaptic terminal. The energy metabolism of the retina is unique in that it is dependent on the light or dark state of the tissue and respiration is more uncoupled from ATP synthesis than in other tissues [31].

RPE cells appear to be different from photoreceptors in that they are specialized to utilize reductive carboxylation as a source of energy [32]. This minimizes RPE glucose consumption for the reduced form of nicotinamide adenine dinucleotide phosphate (NADP(H)) generation, so efficient glucose transport occurs between the choroid and retina. This process is disrupted by excessive oxidative stress and mitochondrial dysfunction, which can be a result of an inability of mitochondria to access necessary substrates, as well as defective electron transport and ATP-synthesis systems [33]. Glycolysis and reductive carboxylation become hindered [32] as pyridine nucleotides, such as reduced nicotinamide adenine dinucleotide (NAD(H)), are depleted, which leads to RPE and retinal degeneration [34,35]. When glycolysis is promoted in these cells, photoreceptors die [36,37], leading to the conclusion that the retina and RPE contribute specific metabolic functions, which maintain their own functional ecosystem [38], both in vivo and in vitro.

### 2.2. Lipofuscin Accumulation in the RPE

Lipofuscin is a lipid-containing, pigmented granule, which accumulates in various tissues throughout the body as a result of aging, and can be found as an accumulation in the RPE [39]. Lipofuscin accumulation is a potential risk factor contributing to the development of AMD [40]. RPE cells convert the condensation product of all-*trans*-retinal and phosphatidylethanolamine to *N*-retinyl-*N*-retinylidene ethanolamine (A2E), which is a major component, and the main chromophore, of lipofuscin [41,42]. There is direct evidence that A2E alters cholesterol metabolism in the RPE, contributing to AMD [43]. In addition, A2E directly causes RPE cytotoxicity by inducing apoptosis through specific inhibition of cytochrome *c* oxidase (COX) [44,45], which leads to the inhibition of oxygen consumption and light homeostasis. This increase in oxidative stress leads to mitochondrial dysfunction, which releases two apoptosis-promoting molecules, cytochrome *c* and apoptosis inducing factor (AIF) [46,47], from the mitochondria of RPE cells [45]. Lipofuscin could, therefore, contribute to the disruption of mitochondrial function and oxidative stress in AMD. This is further highlighted through links that lipofuscin has with the essential trace element, zinc, which is highly concentrated in the RPE. 

In the retina, zinc is required for the metabolism of ingested photoreceptor outer segments (POS) by the RPE [48] and provides protection against oxidative stress [49]. For this reason, it has been suggested that zinc deficiency is linked with AMD, with oral supplementation providing a protective effect [50,51,52]. This decreases the risk of progression from intermediate stages of the disease to the neovascular form in clinical trials [53], but the direct effects of zinc deficiency in the retina remain to be explored extensively. Julien et al. [54] identified an accumulation of lipofuscin and lipofuscin-like products in the RPE of zinc-deficient rats, which may contribute to AMD progression [40]. The authors also proposed that this zinc deficiency contributed to lipofuscin accumulation due to the oxidative stress and functional deficiency of RPE lysosomes as a result of lipid membrane damage caused by lipid peroxidation from zinc deficiency [54]. This ultimately led to incomplete degradation of POS in the dysfunctional RPE lysosomes [55]. This has been further demonstrated in more recent studies exploring the mechanisms of zinc deficiency and supplementation [52,56].

There is also evidence which suggests that the link between lipofuscin accumulation in the RPE and AMD progression is more tenuous. Although A2E is known to be produced in the RPE, it has been shown that, in humans, A2E is preferentially located in the RPE cells of the peripheral retina, rather than the central area of the retina containing the macula [57]. Therefore, in fundus autofluorescence (FAF) imaging, the higher levels of lipofuscin fluorescence associated with the central area of the RPE cannot be exclusively attributed to the concentration of A2E. Other studies have also contradicted the aforementioned results of an increased signal on FAF by quantitatively determining that the fluorescence signal associated with lipofuscin and A2E decreases from subgroups of early to late AMD patients compared to controls [58,59]. As metabolic waste products, including lipofuscin, have previously been associated with AMD progression, clinical trials began to investigate the treatment of GA by limiting the metabolic waste accumulation within the RPE [60,61]. These studies showed no significant reduction in the rate of progression of GA, further highlighting the inconsistent evidence regarding lipofuscin accumulation and its role in the development of AMD.

### 2.3. Sub-RPE Accumulations

#### 2.3.1. Lipid Accumulation

The trafficking and accumulation of lipids and lipid metabolites have long been associated with BrM aging [62,63,64]. As identification of these lipids has progressed, studies identified esterified and unesterified cholesterol (EC and UC, respectively), which are also known as cholesteryl ester (CE) and free cholesterol (FC), respectively [65,66]. Oil red O staining demonstrated that the EC accumulated in the macula seven-fold more than in the periphery of the retina [64,67]. Although EC accumulates exclusively at the BrM, UC and other phospholipids are also found within cellular and intracellular membranes [68]. The association between cholesterol and lipids in the retina, their link to AMD, and potential therapeutic target strategies have previously been reviewed [18,69,70] (Figure 1).

#### 2.3.2. Advanced Glycation End Product Accumulation

Advanced glycation end products (AGEs), oxidized products of non-enzymatic, extracellular protein and lipid glycosylation, have been associated with and implicated in AMD progression for several decades. In one early investigation, the accumulation of these molecules was associated with soft, macular drusen and RPE cells [71]. This accumulation was proposed to contribute to the neovascularization associated with AMD, and more recently, they have been found to accumulate in the BrM [72]. In this localized region, they inhibit protein function and are associated with age-related damage. ARPE-19 cells have also been grown in the presence of AGEs, which leads to an additional increase in the accumulation of lipofuscin, which has its own implication in the pathogenesis of AMD previously discussed.

As these chemical modifications are promoted by smoking, a major risk factor for AMD development [73], it is not surprising that AGEs and their receptors (RAGEs) are suggested to promote the development of AMD. A more recent study investigated the activation of the AGE receptor (RAGE) when the threshold for AGE accumulation is reached [74]. The activation of RAGE is associated with transitioning an acute inflammatory response to a more chronic disease, as indicated in this study, which demonstrated that RAGE was significantly associated with CNV in mice.

#### 2.3.3. Drusen Accumulation and Development

One of the main issues surrounding the prevention of AMD is that the early stages of the disease are often asymptomatic. This means that AMD is often identified when a patient has already progressed to the intermediate stage of the disease and might be suffering from partial sight loss. Early stage AMD can still be characterized by the presence of sub-RPE deposits, which accumulate with age and are associated with the thickening of Bruch’s membrane. Sub-RPE deposits are focal, termed drusen, or diffuse, termed basal laminar (BLamD) or linear (BLinD) deposits. Their formation can contribute to RPE detachment and photoreceptor death [75,76,77]. Lipids, proteins, minerals, and cellular debris are constituents of all sub-RPE deposits, but the composition of the various types can differ [78]. Hard drusen are associated with the normal aging process while intermediate and large soft drusen, as well as BLinD and BLamD, are implicated as contributing to increased disease susceptibility, especially when they are present in the macula [79]. Using new clinical imaging modalities, sub-RPE deposits are becoming better phenotyped in clinical settings [80] and, in combination with laboratory imaging, can help to develop better diagnosis [81].

Although drusen form in the natural aging process, when they increase in number and size, they are critical for the development of late-stage AMD. Indeed, an increase in drusen volume, as measured with spectral domain optical coherence tomography (SD-OCT) in vivo, has been shown to increase the risk of progression from the intermediate stages to advanced AMD [82,83]. Various research groups have begun assessing the molecular constituents contributing to this pathological accumulation. The molecular composition of drusen has been studied in the past, with a major focus on the proteins present in drusen. Common components repeatedly found include apolipoprotein E [84]; amyloid components, including amyloid β [85,86]; complement components; and vitronectin [87,88]. Although the protein composition of sub-RPE deposits is not limited to these proteins, it is worth noting that they were also found to be coating hydroxyapatite (HAP or Ca_5_(PO_4_)_3_OH) spherules [89] and large HAP nodules [81]. These studies suggest that metabolic changes associated with mineral formation are involved. It has been shown that HAP can be deposited onto cholesterol-containing extracellular lipid droplets, after which proteins can bind and oligomerize to start forming the sub-RPE deposits (Figure 2). This HAP deposition and subsequent loss of permeability of the BrM can be modelled in primary cell culture [90], therefore, the metabolic changes associated with mineralization can now be explored.

#### 2.3.4. Trace Metal Homeostasis

Altered metal ion homeostasis has also been implicated in AMD. Apart from zinc and calcium, which were discussed in Section 2.2. and Section 2.3.3., respectively, another metal ion commonly implicated in AMD is iron. 

Iron is well known to contribute to retinal degeneration [91], where it causes damage through the induction of oxidative stress. Within the RPE, the accumulation of iron contributes to the buildup of lipofuscin [92]. In the sub-RPE space, iron accumulation can interfere with molecular pathways, such as the complement system [93]. 

When the metal ion content of sub-RPE deposits were determined in the macula, equator, and far periphery of the retina [94], iron appeared to have the lowest concentration when compared to zinc or calcium, regardless of the geographical locations. 

While the underlying mechanisms for the positive effects of zinc are not yet fully understood, recent evidence suggests that there are multiple effects on the RPE of externally added zinc [52]. However, uncontrolled regulation of zinc levels can have negative effects at both ends of the spectrum. Zinc deficiency appears to be linked to lipofuscin accumulation in RPE cells [54] and it has been shown to impair the phagocytic and lysosomal activity of RPE cells through lipid peroxidation [56]. Alternatively, an excess of zinc contributes to RPE cytotoxicity [95] and can lead to zinc deposition, notably in the sub-RPE space [96]. There is also evidence suggesting that zinc is involved in the oligomerization of CFH and the modulation of the complement cascade and contributes to the protein content of sub-RPE deposits [93].

### 2.4. Choroid-BrM-RPE Interaction

The choroid is the vascularized layer of the eye, which is located between the retina and the sclera, with the innermost layer (choriocapillaris) located on the basal side of the BrM. The retina of humans and other non-human primates is supported by the underlying choroidal vasculature and the retinal vasculature, which supports the inner retina. These dense capillaries are fenestrated on the retinal surface to supply oxygen and nutrients to the RPE and photoreceptors, as well as removing waste products generated in these highly metabolic cells. Studies on the blood flow of the choroid have been carried out [97], where it has previously been determined that a small portion of the choroidal blood flow reaches the retinal photoreceptors. This is primarily due to the high blood flow relative to the small tissue mass [98,99], which exists to account for the poor (< 1 volume %) oxygen extraction from the blood to supply the outer retinal structures [100,101]. The high metabolic demands of the photoreceptor inner segments means that, under normal conditions, there is a large oxygen supply from the choroid to help overcome the issue of distance [102]. This causes an issue when there is any disruption to the distance between the choroid and photoreceptors (such as in the presence of sub-RPE drusen), as the lack of oxygen will lead to progressive photoreceptor degeneration. Chirco et al. [103] have recently reviewed the changes that occur to the choroid with aging, at both a structural and molecular level, and how this relates to AMD disease progression.

The blood-retinal barrier formed by the tight junctions of the RPE is important for the supply of nutrients to the photoreceptors. The BrM was first thought to form part of this barrier, but it became apparent in early studies that it was relatively permeable compared to the RPE and, instead, was more important in the removal of waste products from the retina. Unfortunately, as the BrM thickens with age, it becomes increasingly impermeable to the low concentrations of waste products that begin to accumulate in the sub-RPE space. The opposite has been found to be the case for the choroid, which appears to thin with age, particularly at the fovea [104]. This effect happens alongside a decrease in von Willebrand factor and human leukocyte antigen (HLA) class I proteins [105], both vascular specific proteins. This suggests that there is a dedifferentiation of the endothelial cells in eyes with early AMD and, along with the loss of the RPE cell layer, provides an insight into the steady metabolic dysregulation associated with the development of AMD.

## 3. Metabolomics in AMD

### 3.1. Introduction into Metabolomics

Metabolomics can be defined as the measurement of all small molecule metabolites within a biological system [106], including those that are environmental in origin. There is an increasing understanding of the components associated with the transcriptome and proteome, but the metabolome offers an integrated perspective of cellular processes and the effect environmental factors may have on the biological state (Figure 3). Compared with the other ‘-omics’ approaches for investigating changes in the physiology of individuals and populations [107], metabolomics remains in its infancy. However, its popularity has rapidly increased over the last two decades. Mass spectrometry (MS) and nuclear magnetic resonance (NMR) spectroscopy are the predominant platforms that are used to obtain metabolomic profiles from a diverse range of sample types. To increase its resolution and sensitivity, MS is combined with different separation techniques, such as gas chromatography (GC) and high-performance liquid chromatography (HPLC) [108]. NMR spectroscopy is useful for metabolomic investigations because of its reliability and for the structural information it provides. NMR is not as sensitive as MS, making it more useful for quantifying high abundance metabolites. Furthermore, NMR does not always offer the high-throughput data acquisition that is common with mass spectrometers. However, there is evidence that this barrier is now being overcome [109].

Sample preparation is critically important in metabolomic investigations because metabolites could be introduced, altered, or removed during processing, which may not reflect the actual molecular state of an organism. For this reason, there is a preference for the use of biofluid samples over tissues or cells for some applications because they require less processing [111,112,113]. As the biofluids are in contact with various organs throughout the body, they also provide a more representative global metabolic profile. Even with the best sample preparation, there are still limitations as the current techniques of subsequent separation and detection methods are unable to identify all in vivo metabolites, primarily due to the heterogeneous and diverse chemistry of the currently known metabolites [114,115]. As these separation and detection platforms improve, there will be increasing numbers of studies investigating greater depths of metabolomics knowledge and understanding, as already evidenced by the rapid expansion of identified human metabolites from just over 6,800 to over 110,000 metabolite entities [116,117].

### 3.2. Retinal Tissues

In AMD, metabolomics studies have been conducted using samples obtained from animal models and human patients. A summary of untargeted cellular, human, and animal AMD metabolomics studies and the specific techniques used can be found in Table 1. One such study explored the metabolic changes associated with photoreceptor degeneration, a consequence of late stage AMD and GA, as well as whether induced pluripotent stem cell (iPSC)-generated RPE cell grafts altered this metabolomic profile [118]. The metabolites from rat eyes at different time points post-graft were processed by HPLC-QTOF-MS (quadrupole time-of-flight mass spectrometry) and were analyzed against the METLIN database. When comparing 52-week old dystrophic rats against age-matched control rats, significant changes were reportedly observed at 3 weeks, with most changes occurring at 52 weeks. Of the 203 metabolites which significantly changed in the diseased samples, more than half were phospholipids and various oxidized species. Glycerophosphocholines were the most abundant subclass that changed, followed by glycerophosphoethanolamines, long-chain acylcarnitines, monoglycerols, fatty acid amides, and long-chain polyunsaturated fatty acids (LC-PUFAs). There was a general increase in the fold-change of phospholipids in the dystrophic tissue, suggesting a higher level of lipid metabolism in the diseased eyes. Alternatively, there appeared to be a downregulation of the acylcarnitines, which are important for energy homeostasis and RPE function, and of the docosahexaenoic acid ω-3 fatty acid (DHA), which is necessary for retinal homeostasis [119,120]. DHA is the precursor of various phospholipids found enriched in the outer segment membranes of the retina, including lysophosphatidylcholine (lysoPC), lysophosphatidylethanolamine (lysoPE), and lysophosphatidylserine (lysoPS), all of which were downregulated in a similar manner as DHA.

A similar decrease was observed over time for all-*trans*-retinal (atRAL) in Royal College of Surgeons (RCS) rats, which is indicative of photoreceptor degeneration [121], along with an increase in the toxic A2E fluorophore. This resulted from inefficient clearance of atRAL in the dysregulated visual cycle [122]. The final part of their study identified that stem cell-derived RPE rescued the loss of DHA-lipids associated with the diseased phenotype, suggesting RPE transplantation could be a metabolic mediator with therapeutic benefits [123].

Comparable results have been obtained when using mouse models of photoreceptor degeneration to compare healthy mice and AMD patients. Orban et al. [124] used tandem MS to analyze the *Abca4^-/-^ Rdh8^-/-^* mice, which exhibit retinal degeneration under light-induced photoreceptor damage. This revealed that 11-*cis*-retinal and DHA levels were both decreased, whilst intense light produced increased levels of prostaglandin G2. The animal model metabolite differences also appeared in serum from AMD patients suffering from the nonexudative form of the disease in the dysregulation of DHA. Intense light also reduced the levels of DHA in wild-type mice, but this did not lead to photoreceptor degeneration. One of the differences between the disease seen in mice and the AMD patients was the statistically significant increase in arachidonic acid (AA) in the human serum, which was not present in the mice eyes. This study also employed a statistical model as a predictive tool for the identification of AMD, which gave a 74% chance of identifying AMD patients when compared to controls using only DHA and AA. The inclusion of the AMD-associated Ala^69^-Ser variant in the *ARMS2* gene did not significantly improve this model. This was primarily because only a small correlation was found between the genetic mutation and AMD, meaning that, in these samples, the genetic screen for this amino acid transition was not a good predictor of AMD development. This variant has previously been shown to be associated with AMD progression and could be used as a predictive tool [11], which indicates using multiple models with independent biomarkers are a better diagnostic strategy to employ.

As it is well documented that the environment can impact the metabolome, it may also be worth noting how different tissue collection methods are employed for metabolomics studies. One study has highlighted the potential impact of anesthesia and euthanasia on metabolomics studies carried out on tissues [132]. Here, it is highlighted that there are tissue-specific changes dependent on the method of tissue sampling. For example, after euthanasia, skeletal muscles showed higher levels of glucose-6-phosphate, whereas nucleotide and purine derived metabolites accumulated in the heart and liver, when compared to anesthetized animal tissues. Although the authors recommend utilizing anesthesia for tissue collection, this may not be feasible for every study and, instead, highlights a point of consideration in tissue metabolomics studies. 

Another consideration for metabolomics studies is post mortem time, especially in humans. There is evidence from GC-MS and ultra-high performance LC-MS (UHPLC-MS) studies in the eye that statistically significant post-mortem changes can be observed in a number of metabolites, although most metabolites were stable for up to eight hours post-mortem [133]. 

### 3.3. RPE Cells

Although several cellular models have been developed to replicate the RPE layer in vivo, the complex cellular environment of the retina means the exact environment is difficult to mimic. The human fetal RPE (hfRPE) cellular model is possibly the most reliable model of RPE cell function as it appears to replicate the physiological RPE morphology that other cell models and cell lines do not [134,135,136]. The use of this cell model as an accurate indicator of cellular metabolism may be helpful for metabolomics investigations [137]. The retina contains many different cell types, which have preferential metabolic pathways for energy metabolism, such as aerobic glycolysis in photoreceptors and reductive carboxylation in the RPE. In addition to the metabolism studies mentioned previously [23,125,138], these primary cell culture models also provide a platform for studying the development of drusen deposits in vivo [139], which are important factors in the development of GA. This has been expanded upon in recent years, with the identification of individual components of the drusen already mentioned [78,89,90]. Of interest is the accumulation of HAP, which is not commonly found in healthy soft tissues. The accumulation of high concentrations of phosphate and calcium, together with other divalent ions, in the sub-RPE space provided a new insight into the development and progression of sub-RPE deposits. It is perhaps unsurprising that there is an accumulation of phosphate and calcium, given that the retina is one of the most metabolically active tissues in the body [140,141,142], and the RPE has a very high calcium content [143]. Sphingolipids are one metabolite class that have been demonstrated to both contribute to, and protect against, photoreceptor apoptosis [144,145,146], as well as potentially having a role in choroidal and retinal neovascularization [147,148]. Another role that could be related to the deposition of calcium-phosphate at the RPE basement membrane is the role sphingolipids and their potential kinases have in calcium mobilization, particularly in response to the influx of calcium via transient receptor potential (TRP) channels [146,149,150,151].

Each of the cell types within the retina must communicate with one another to maintain a regular homeostatic environment and function properly to maintain viability. Recently, Chao et al. [125] investigated the consumption of nutrients and the subsequent transport of metabolites through the RPE cell layer. Through LC-MS/MS, 120 metabolites were identified in the culture medium of hfRPE cells and three were identified as being the most heavily consumed nutrients. Glucose and taurine showed the highest consumption from the apical medium, with proline consumed from both the apical and basal medium after 24 hours. Through isotopic labelling, they were also able to identify that metabolic intermediates from both glucose and proline metabolism were preferentially exported to the apical side of the culture, which is then imported into the neural retina when co-cultured. Proline is an energy source, which is utilized by both the citric acid cycle and the reductive carboxylation pathway. This provides a valuable insight into the unique metabolism that may occur in vivo, demonstrating the use of alternative metabolic pathways in RPE cells and indicating metabolite secretions, which may present as biomarkers that could change when the RPE is in a diseased state. This publication further proved human primary RPE cells are suitable model systems to study the dynamic changes in and around the RPE [125]. Further manipulation of culture conditions will now be able to study differences in the metabolism, which could mimic those found at the photoreceptor/RPE/choroid interface.

### 3.4. RPE Cells and the Retina

The metabolic connection between RPE cells and photoreceptors means that the loss of the RPE cell layer is detrimental, not only to the choriocapillaris, but also as a contribution to photoreceptor degeneration in the late stages of AMD [152,153,154,155]. Knowing that the RPE cells provide a homeostatic platform for the underlying photoreceptors, and that their degeneration contributes to the pathogenesis of AMD, Kurihara et al. [36] investigated the effects of oxidative stress on this cellular interaction as an early indicator of the disease. Using a murine model and primary RPE cells cultured from mice, the authors found that a hypoxic environment, induced from choriocapillaris vasodilation, led to the accumulation of lipid molecules, BrM thickening, RPE hypertrophy, and significant photoreceptor degeneration [36]. In addition to the pathological changes associated in the different cell types, glucose metabolism also became impaired within the hypoxic RPE. This was evidenced by the shift from oxidative phosphorylation to glycolytic metabolic pathways, under hypoxic environments, in vivo. This was supported by increased apical RPE uptake of glucose in vitro, which subsequently reduced available glucose for photoreceptor metabolism. These results are similar to those suggested in other studies looking directly at the effects of RPE and photoreceptor metabolism [125,138]. Through untargeted MS analysis, it was revealed that various forms of acylcarnitines were also significantly different in the von Hippel Lindau (*Vhl*) knock-out mice [36]. When combined with the gene knockout for hypoxia-inducible transcription factor 2 (HIF2α), the levels of acylcarnitines were of a similar level to that in control mice. This is similar to results obtained in other animal models using the same two genetic mutations [156,157], which suggests that HIF2α could play a crucial role in retinal lipid regulation as the combined genetic knockout of *Vhl*/*Hif2α* partially restores the wild-type lipid homeostasis. This is just one example that investigated the effects of oxidative stress on the RPE, but the overall literature in oxidative stress and the RPE were recently reviewed in the context of neovascular AMD (NVAMD) [158].

### 3.5. Biofluids

One of the first investigations into the metabolic changes associated with oxidative stress in AMD was performed in blood, but later studies on different biofluid metabolites were also investigated. These are summarized in Table 2. In the first study, the authors targeted metabolites, which were products of thiol redox reactions and lipid peroxidation [159]. Blood plasma samples from AMD patients and control individuals were assessed for cysteine (Cys), cystine (CySS), glutathione (GSH), isofurans (IsoFs), and F_2_-isoprostanes (F_2_-IsoPs), chosen based on previous results demonstrating that increased oxidation of these metabolites were associated with AMD risk factors [160,161,162]. Using HPLC and gas chromatography MS (GC-MS), they identified that only the levels of CySS were significantly higher (9.1%) when comparing AMD patients and controls. The levels of CySS were also significantly greater between neovascular and advanced AMD patients compared to controls. However, when the analysis was adjusted for age, gender, and smoking, the results were no longer significant and so the results are only suggestive of possible systemic metabolite changes.

An additional group used a targeted approach to investigate the association of long-chain ω-3 PUFAs, triglycerides, and high (HDL) and low (LDL) density lipoprotein-cholesterols in patients with NVAMD [163]. Serum and red blood cell membrane (RBCM) fatty acids were determined by GC, whereas enzymatic colorimetric and electrophoretic methods were employed to measure triglycerides and the serum lipoprotein-cholesterols. Compared to control samples from individuals with no history of ocular diseases, the NVAMD patients had significantly lower plasma triglycerides, serum eicosapentaenoic acid (EPA), RBCM EPA, and DHA after adjustment for age and sex. Similar levels of plasma total, HDL-, and LDL-cholesterol, as well as serum DHA and EPA+DHA (omega-3 index), were observed in both NVAMD patients and the controls. These results build on a previous population-based study by the same group [164], which showed a trend of a decreased risk to progress to NVAMD in patients with higher plasma ω-3 LC-PUFAs. A decrease was also found between the plasma EPA in the Alienor study [164], which was not associated with AMD, as opposed to the significant association of AMD with serum EPA in the more recent study [163]. Decreased levels of DHA in association with AMD has previously been highlighted in Section 3.2. [118,124]. 

The same group that investigated oxidative stress in AMD patients [159] progressed to an untargeted metabolomic investigation of plasma in NVAMD patients and age-matched controls with no clinical signs of AMD [126]. Using LC-FTMS (Fourier-transform mass spectrometry), there appeared to be significant changes in the intensity of 94 metabolites between the two cohorts. A more detailed analysis revealed a total of 40 metabolites that overlapped between the log2 transformed and non-transformed data. Cluster analysis of these 40 metabolites showed significant increases in certain peptides, modified amino acids, and natural products, defined as either a metabolite synthesized by a living organism or a metabolite specifically involved in secondary or specialized metabolism [165]. Significant decreases were found in bile acids, vitamin D-related metabolites, and dipeptides (histidine-arginine and tryptophan-phenylalanine) in NVAMD patients compared to controls. Pathway analysis of these metabolites revealed that phenylalanine and dopaquinone in the tyrosine metabolism pathway, which aids in the synthesis of melanin [166], or aspartate and glutamine in the urea cycle pathway, a natural process to remove excess nitrogen from the body, are involved.

Similarly, Luo et al. [127] assessed the plasma metabolomics profile of a Chinese cohort of patients suffering from NVAMD and healthy controls using UHPLC-QTOF MS. In this study, there were 10 metabolites that differed significantly between the two groups, the majority of which were amino acids, and the most significant finding was an increase in l-phenylalanine in AMD patients. A further analysis of the associated metabolic pathways revealed that most metabolites belonged to the amino acid biosynthesis pathway. The most common metabolites found across the metabolic pathways had also been previously shown as having a significant change by Osborn and colleagues [126].

Untargeted metabolomic studies assessing metabolic differences between healthy individuals and AMD patients are increasing in frequency. One such study used LC-MS/MS analysis to assess the blood serum of 60 individuals characterized as either healthy controls, patients suffering with CNV, or patients suffering with polypoidal choroidal vasculopathy (PCV) [167], a subtype of AMD commonly found in Asian populations [170]. It was shown that glycerophospholipids, amino acids, di/tripeptides, ω-3 and -6 PUFAs, and various carnitine species were all elevated in both CNV and PCV patient samples. In total, there were 197 significantly altered metabolites across both conditions compared to serum metabolites from controls. Only one metabolite (pinolenic acid) was shown to be different between CNV and PCV patients, which suggests there could be significant metabolic overlap between these two diseases [170]. The results from this study provided additional knowledge to previous research, which compared the serum lipid profiles of PCV patients and compared these to controls [171]. A total of 41 metabolites were significantly altered in PCV patients, which included increases in 18 phosphatidylcholines (PCs), eight sphingomyelins (SMs), three lysoPCs, three platelet-activating factors (PAFs), one lysophosphatidic acid (LPAs), and one phytosphingosine. Significant decreases were found in one PC, three LPAs, two sphingosines, and one phosphatidylethanolamine (PE) [171].

The results from the PCV and AMD comparison study [170] were expanded in 2017, where investigations into the plasma metabolomics profiles of AMD patients and healthy age-matched controls were conducted using NMR [128] and UPLC-MS analysis [129]. For the plasma samples analyzed by NMR, two large cohorts of AMD patients were recruited at two separate study locations (Coimbra and Boston) and were characterized by the severity of their AMD progression. These were compared against each other, as well as being compared against control samples from both study cohorts. There were noticeable differences in the metabolic profiles between AMD severity stages, including between samples from early-stage AMD and the control groups. There were higher levels of circulating creatine, acetate, dimethyl sulfone, C18 cholesterol, and high-density lipoprotein (HDL)-choline, whilst there were lower levels of unsaturated fatty acids between controls and the early AMD Coimbra cohort [128]. Across the AMD severity stages, there were minor differences in low-molecular weight (M_w_) metabolites, such as higher pyruvate for intermediate AMD and lower levels of histidine, acetoacetate, and β-hydroxybutyrate for late AMD. Samples from the Boston cohort showed slight differences in the low-M_w-_ metabolites, with higher and lower levels of glutamine for early and intermediate AMD, respectively. Lower histidine levels were found in intermediate and late AMD, with lower levels of alanine in late AMD. As evidenced by the metabolites found between the two different cohorts, there appears to be a geographic influence on the metabolites that present as significantly different in AMD patients, again highlighting the influence of the environment on metabolomics studies and disease profile.

A subsequent study by the same group analyzed plasma metabolites from patients with different stages of AMD progression and age-matched controls, using UHPLC-MS [129]. Here, 87 metabolites were identified as differing between AMD and controls. Most of the metabolites were members of the lipid super-pathway (82.8%), followed by amino acids (5.7%). Similarly, six out of seven of the most significantly different metabolites were lipids, the exception being adenosine. In terms of increasing AMD severity groups, 48 metabolites significantly differed and all but one of the most significant metabolites were involved in lipid pathways. Pathway analysis demonstrated that most were involved in glycerophospholipid metabolism. It is important to note that Osborn et al. [126] were unable to distinguish the lipids that were present in their samples and so they were not analyzed and cannot be compared to the study by Laíns et al [129].

Recently, the limitations of the original paper by Osborn et al. [126] have been improved on in a study, which investigated the metabolites and associated metabolic pathways of a larger cohort of NVAMD patients [131]. Plasma samples were collected from NVAMD patients, who exhibited extensive CNV, subretinal hemorrhaging or fibrosis, or photocoagulation scarring in one or both eyes. Control individuals were identified as having fewer than 10 small drusen and no macular pigment changes in both eyes. Untargeted metabolomics was carried out using LC-MS, which identified 10,917 unique metabolite features. Analysis of these features highlighted 159 metabolites that were distinguishable between NVAMD patients and controls. There was an increase in 110 of these metabolites in NVAMD patients, with 49 showing decreased levels, compared to controls. Further analysis identified 39 metabolites with medium to high confidence. Only metabolites that have been exclusively identified have been listed in Table 2. Acylcarnities, amino acids, bile acids, lysophospholipids, and phospholipids were amongst those annotated and, following Bonferroni-corrected pathway analysis, the carnitine shuttle pathway was revealed to be significantly altered in NVAMD patients. Further LC-MS/MS analysis confirmed the identity of five of the six carnitine shuttle pathway metabolites, which all showed a significant increase in NVAMD patients. This larger cohort study builds on previous NVAMD metabolomics research, which have also identified acylcarnitine and bile acid alterations in the patient population [126,127].

Preliminary studies published as abstracts on metabolite profiles in AMD have also been compared using NMR metabolite profiles of laser-induced mouse models of NVAMD and human AMD patients, identifying lactate as an important metabolite in both cases [172,173]. Preliminary studies in urine have also been carried out [168,169]. Here, the NMR profiles for NVAMD cluster well, whereas the clustering of the nonexudative AMD patients was more diffuse. This may suggest a more heterogeneous population and highlights differences between AMD subgroups. In NVAMD patients, there were notable increases in arginine and decreases in glucose, lactate, glutamine, and glutathione. Results obtained in urine showed an overlap between neovascular and nonexudative AMD, suggesting that the two forms of the disease could be linked [168,169]. The metabolomics profile of subretinal fluid has also recently been investigated [174], in the context of other ocular diseases, with 651 metabolites identified in a small sample size of three patients. 

It is perhaps not surprising that lipids are the most consistent metabolites that show changes in the different stages of AMD. The role that lipids play in the pathogenesis of AMD is becoming clearer though it is still to be fully elucidated [175]. The most significantly associated metabolites in these studies belonged to the glycerophospholipid family, which provide structural stability and fluidity to neural membranes. The most likely source of these are the degrading cell membranes and photoreceptor outer segments’ discs. Although these studies provide a basis for the identity of potential metabolic biomarkers present in the circulation and tissues of AMD patients and models of AMD disease states, they may not be directly relatable to the process occurring in the retina. 

## 4. Alternative Approaches for AMD Metabolomics Studies

### 4.1. Tears

Tears are a biofluid originating from the anterior of the eye and are a potential source of metabolite biomarkers directly linked to ophthalmology. Although relatively small in volume [176], technological advancements now make it possible to characterize the proteomic [177,178], lipidomic [179], and metabolomic [180] composition of the human tear. As a source of biomarkers, the tear has been used to assess the disease profiles of various ocular diseases, including, but not limited to, dry eye disease, keratoconus, trachoma, and diabetic retinopathy, which have been reviewed elsewhere [181]. The majority of tear analysis studies have focused on the proteome as the relative amount of protein is greater than metabolites. Along with this, a single technique is unable detect all tear metabolites due to no standardized collection, analysis, or identification methods. Despite this issue and not being in direct contact with the retina, tears provide a non-invasive source of metabolomic biomarkers [182]. 

One of the first exclusive characterizations of the human tear metabolome aimed to use a standard clinical method of tear collection and to develop an analytical platform, which could be applied to characterize the global repertoire of human tear metabolites [180]. In this study, tears of healthy individuals were collected using the clinically utilized Schirmer strips, separated by UFLC (ultra-fast LC) and analyzed by Q-TOF MS/MS. This untargeted method of metabolite analysis identified 60 metabolites from a range of 16 compound classes. Of these metabolites, 44 were ‘novel’ as they did not correspond with metabolites identified in previous targeted studies (see Table 1 in [180]). This set the precedent for what could be achieved in tear metabolome studies, but it was clear that this method did not measure some well-known metabolites (e.g., measurement of glucose and ascorbic acid was affected by background interference). 

Very few lipid species were identified, but within the literature, others have identified several classes of lipids using targeted analysis. These include free cholesterol [183], phosphatidylcholines [183,184], SMs [183,184], wax esters [183,185], lysoPC [186,187], triacylglycerides, ceramides, and phosphatidylethanolamines [186]. A further study investigated lipid composition during collection with Schirmer strips using untargeted analysis [179]. Tears were collected either by capillary tube or Schirmer strip and extracted lipids were analyzed using HPLC-MS. Over 600 lipid species across 17 lipid classes were detected, the majority of which were categorized as either wax esters or cholesteryl esters.

Contact of parts of the strip with the eye [188], particularly the meibum [189], clearly contributes a large proportion of free cholesterols, sphingolipids, and phospholipids, and therefore the tear lipidome is less complex than the meibum lipidome. These results were consistent with previous studies investigating lipid classes in human tears, but this was the first extensive characterization of the tear lipidome, where novel metabolites were also identified [179]. This includes cholesteryl sulfates, which, as a stabilizing agent [190], could contribute to the amphiphilic sublayer of the tear film. This study indicates the limitations of tear collection methodologies within the context of metabolomic investigations. Schirmer strip collections yield the highest absolute amounts of lipids and are routinely employed in the clinic. Although the level of background noise was high in blank Schirmer strips, no endogenous tear lipids were detected [179]. Interestingly, the strips act as a chromatographic system for lipid metabolites [179,186]. The aqueous fraction of the tear travels further along the strip than the non-polar lipids. The strips can capture an accurate representation of the lipidomic profile of tears and their relative concentrations when compared to spiking with artificial tear solutions [179]. It should be noted that tears used in this study were obtained from patients with dry eye syndrome [179] and so their metabolomic profiles and relative metabolite concentrations may differ from other patient groups.

Tear glucose levels have been considered as a non-invasive method of detecting the early stages of diabetes [191] and they are increasingly being investigated for their use as a sensor for diabetes mellitus [192]. It is well known that tear glucose levels are variable diurnally [193] and from which eye the sample is taken [192]. However, there is evidence suggesting that the tear glucose levels are reflective of blood glucose levels using enzyme-based and amperometric biosensors [194,195], suggesting future clinical investigations of tear metabolites are worthwhile. However, there have been questions raised as to the reliability of the results obtained and whether they are comparable to blood glucose levels [196,197], and thus needs further investigation. 

Tear samples appear to be a reasonable reservoir of metabolites, their collection is non-invasive, and the samples are easy to handle and cheap to transport, indicating that tear sampling could become a reliable source for metabolic markers for eye as well as other diseases [198]. Saliva has also been studied as a source of potential biofluid metabolites [199,200,201], but there have been no studies investigating the salivary metabolome in eye diseases.

### 4.2. Vitreous and Aqueous Humor

In a similar manner to tears, the vitreous and aqueous humors might be a representative ocular biofluid to use for metabolomics studies in AMD. Although they have to be obtained through invasive procedures, often requiring collection during surgery, they are also being used as surrogate sources of ocular disease biomarkers. Vitreous humor has been studied for metabolite changes in diabetic retinopathy [202,203,204,205,206,207], proliferative vitreoretinopathy [207,208], rhegmatogenous retinal detachment both associated and not associated with choroidal detachment [202,207,208,209], and uveitis [207]. Young et al. [207] investigated the metabolomic profiles of vitreous humor obtained from a variety of inflammatory eye diseases. They were able to demonstrate clear and specific differences in the metabolites obtained from each disease, with a high sensitivity in clinically relevant samples. In a non-clinical context, vitreous humor from sheep, pigs, and rabbits was profiled using targeted methods [210]. This revealed that acetylcholine esterase activity varied across species, but less so between breeds of rabbit. Untargeted LC-MS analysis also found differences in metabolite profiles that may simply reflect the diets of each animal, and this may have relevance to studies in humans. However, it should be noted that, based on a study on rats, only 1.6% of the total metabolic profile overlapped between the vitreous and the retina [133] although this might change in pathological states. 

Aqueous humor has also been explored as a source of metabolite biomarkers in different diseases, but the number of studies is more limited. In an acute model, glaucoma changes in glucose and citrate levels were detected [211]. In a chronic glaucoma mice model, an increase of sphingolipid and ceramide species was found [212]. Furthermore, in a chronic rat model for glaucoma, increases in acetoacetate, citrate, and various amino acids, including alanine, lysine, and valine, as well as a decrease in glucose levels, were found using ^1^H-NMR [213]. For acute and chronic glaucoma studies in human aqueous, humor phospholipids [214], cholesterol [215], sphingolipid, and ceramide species [215,216] profiles have been demonstrated to be dysregulated. 

Aqueous humor has also been used in the identification of metabolic changes associated with myopia [217,218]. Using a dual platform of capillary electrophoresis–mass spectrometry (CE-MS) and LC-MS, one of the studies identified 40 metabolites [217]. Of these, 20 were deemed to be significantly different between varying stages of myopia. Increases in arginine, citrulline, and sphinganine were associated with high myopia while increases in aminoundecanoic acid and dihydro-retinoic acid were associated with low myopia [217]. The other study used GC/TOF-MS and compared the metabolites present in the aqueous humor of patients with high myopia and compared these to controls [218]. A total of 242 metabolites were identified, with significant increases observed in 27 and significant decreases in two metabolites [218]. 

Metabolomic profiles in human aqueous humor have been compared with serum metabolites from the same patients [219]. The most notable metabolite to differ between the two biofluids was ascorbate, attributed to the ascorbate-specific pumps at the blood-aqueous border. Other differences were attributed to the differential metabolic activity of the compartmentalized ocular tissues, but the potential for post-mortem artefacts has also been raised [219]. Due to the invasiveness of obtaining vitreous and/or aqueous humor, their use will probably be limited to those where surgical intervention is a necessity.

### 4.3. In vivo Imaging

A different approach to obtaining metabolomic information, and of interest for retinal studies, is the use of in vivo fluorescence imaging. Imaging approaches offer the advantages of data collection with high spatial resolution, modest intervention, high safety, and low cost of data. Due to their accessibility, the eye, and retina in particular, are well suited for optical studies compared to most organs. This has fueled the development of numerous approaches for diagnosis and therapy in the eye. Many metabolites highlighted in Table 1, and in previous sections, have little or no fluorescence emission at wavelengths longer than 300 nm, nor unique IR nor Raman spectra, limiting the scope of fluorescence imaging as a broad metabolomic approach. However, the nicotinamide adenine dinucleotide coenzyme NADP(H), and to a lesser extent flavin mononucleotide (FMN), flavin adenine dinucleotide (FAD), and pyridoxal/pyridoxamine, exhibit useful visible fluorescence under certain conditions [220]. These molecules are also intimately linked to numerous metabolic pathways and offer information about metabolite fluxes through those pathways. Of interest are the nicotinamide adenine dinucleotides, NAD(H) and NADP(H), which are the principal carriers of reducing equivalents within cells. NAD(H) is crucial for energy production in the cell as it is the primary transporter of electrons for oxidative phosphorylation, whereas NADP(H) provides the reducing equivalents for the neutralization of cellular reactive oxygen species (ROS). The importance of NAD(H) to the energetic state of cells was recognized decades ago, where the proportions of NAD(H) and NADH were measured spectrophotometrically and by fluorescence [221]. Naturally, as a cell acquires energy through different routes (oxidative phosphorylation, glycolysis, fermentation, reductive carboxylation) and from different fuels (glucose, fatty acids, ketone bodies), different pathways are activated or deactivated, and the broader metabolome will reflect this.

Recently, more powerful techniques have been developed to study NADP(H) and other fluorophores. For example, the fluorescence emission spectra of NAD(H) and NADP(H), free in solution, are very similar (a broad peak around 460 nm after excitation near 340 nm) and so the proportions of each cannot be distinguished based on spectra alone. However, the reduced forms of both exhibit different fluorescence lifetimes when free in solution and bound to proteins [222,223]. The fluorescence lifetime is the average amount of time the fluorophore spends in the excited state between excitation and emission, and is typically in the range of nanoseconds [224]. This can either be measured in the time domain with a technique known as time correlated single photon counting (TCSPC) [225], or in the frequency domain by phase fluorometry [224]. While the lifetime is a general property of each fluorophore, for some fluorophores, the lifetime is very sensitive to environmental conditions. When free in solution, NAD(H) has a low quantum yield and a mixture of fluorescence lifetimes between 0.3 and 0.8 nsec. When protein-bound, the lifetime ranges from 1.5 to 6 nsec. The development of fluorescence lifetime-based imaging microscopy (FLIM) [224,225] made it possible to collect specimen images where the contrast comes from differences in lifetime, not intensity. Studies have appeared demonstrating that FLIM can distinguish between free and protein-bound NAD(H) [226] and between intracellular NAD(H) and NADP(H) [227], a result not possible when only detecting their spectrally identical fluorescence [228]. The metabolic state and composition of a tissue can also be assessed by measuring the autofluorescence lifetime of endogenous fluorophores [229,230,231,232], including protein-bound NADP(H) [223].

In in vitro studies, FLIM has been utilized to assess different structures within RPE cells [233], as well as in retinal tissue from donors with AMD [234]. Miura et al. [233] identified that the fluorescence lifetime observed inside and surrounding cultured RPE cells increases significantly when oxidative stress is induced. This enabled the discrimination between the granules associated with increases in metabolic stress from the melanosomes seen under normal conditions. From retinal tissues [234], the RPE and BrM could be discriminated from one another using their fluorescence lifetimes. This was primarily because of the presence of lipofuscin in the RPE, which had a shorter lifetime than other emitters. Where it was present, drusen could also be discriminated from the RPE and the BrM. In addition, it was found that different drusen had different lifetime distributions, indicating that different fluorophores, or the same fluorophores in different environments, contribute to the FLIM image [234]. Schweitzer et al. [234] also found lipofuscin-like FLIM signatures in structures within drusen, which they interpreted as indicating a role of lipofuscin in the formation of sub-RPE deposits. 

The analysis of the ocular fundus has been extended to include fluorescence lifetime imaging ophthalmoscopy (FLIO), a technique analogous to FLIM [235]. Fluorophores that are interesting for AMD are redox coenzymes, NAD(H) and FAD, as well as lipofuscin [236], AGEs [71], and collagen. The excitation and emission spectra of these endogenous fundus fluorophores have been investigated in vivo, which showed that it would not be possible to distinguish them based on autofluoresecnce [231]. However, changes in fluorescence lifetime, especially when the metabolic environment is altered, could identify novel signatures that are associated with AMD [231]. When performing FLIO, it is worth considering the strong fluorescence of the lens [237], which cannot be entirely suppressed, but which can be overcome using specific analysis software [238]. Multiphoton excitation may help minimize such background fluorescence, but the safety considerations should be made when utilizing focused, high peak power picosecond lasers for in vivo studies [239,240].

Following on from the in vitro observations described by Miura et al. [233], the in vivo FLIO has also been utilized as a tool to investigate the effects of induced oxidative stress on RPE degeneration and photoreceptor loss in a mouse model [241]. Following intravenous sodium iodate injection, which induces RPE degeneration, the retinal autofluorescence lifetimes increased over a 28-day period compared to control mice. In contrast, intraperitoneal injection of *N*-methyl-*N*-nitrosourea, which causes the specific degeneration of the photoreceptors, resulted in shorter lifetimes being observed [241]. From these results, the authors suggested that short lifetimes are present in the RPE, but are altered with the overlying retinal structures. FLIO imaging has also been used to distinguish AMD patients from controls [242]. It was found that images from AMD eyes had a significantly longer retinal fluorescence lifetime, and they found that the lifetimes may vary from drusen to drusen, although distinguishing drusen on FLIO images appears to be challenging [242]. Similar observations were made in another study [243]. However, lifetime changes are present in 36% of the healthy controls, which suggests these observations could be the result of aging [243]. Whether these fluorescence lifetime changes are associated with the metabolomics changes mentioned above will need to be investigated further.

The ability to distinguish between drusen subtypes and different structures in the retina is complementary to the multimodal imaging approaches described previously [81,244]. These studies indicate that fluorescence lifetime imaging approaches (FLIM and FLIO) could become tools for studying not only the biology and metabolomics of the retina, but also the pathologic processes that lead to diseases, such as AMD. Several comprehensive reviews detailing the principles of fluorescence lifetime imaging and its application as a clinical tool has recently been published [245,246]. 

## 5. Conclusions

In summary, the complex repertoire of metabolites found in the retina is only beginning to be revealed, and there is scope to identify those associated with the development of AMD. Clearly there is a dysregulation of lipids in patients suffering from AMD, and various models exist for investigating AMD biomarkers. Considering the close association between photoreceptor degeneration, RPE health, sub-RPE deposit formation, and choroidal changes, it is perhaps not surprising to find lipids and molecules associated with metabolic fluxes in the studies highlighted. Plasma metabolomics could be a convenient tool for analyzing a wide range of factors potentially contributing to AMD, however, the number of published studies compared to other ‘-omics’ methodologies still remains low [110]. Further efforts are clearly needed here whilst due consideration of the blood-retinal barrier is required. It is still unclear whether the systemic biomarkers identified in biofluids can be representative of the changes in the eye. Alternative biofluids, such as tear, saliva, and vitreous, are also viable possibilities for detecting local and systemic changes in AMD patients. 

Some of the current barriers associated with large-scale epidemiological metabolite studies are being overcome by advances in technology for untargeted metabolite analysis [247]. Comprehensive study design is crucial for high accuracy and good quality metabolomics data collection, which means other potential limitations include, but are not limited to, subject selection, sample selection, collection, handling, storage, and preparation [21,248]. As part of overcoming these limitations, it will become necessary to establish reference lists of metabolites obtained from various biofluids in larger cohorts of healthy, age-similar patients. Similar cell-based libraries are being developed for proteomic integration [249]. For metabolomics, this will require additional advances in the analytical technologies, as well as strong collaborations to systematically ensure a standardized benchmark can be obtained.

The introduction and application of lifetime based clinical imaging with FLIO now allows a non-invasive approach for the identification of metabolic changes in the retina. This is a new methodology that needs further validation, but the potential to use this approach for clinical practice is very appealing. Apart from the imaging of endogenous fluorophores, there is a possibility to deliver markers that change their fluorescent lifetime once they are bound to specific metabolites. While this approach is still in its infancy [250], once proven to be safe, it could provide new insight into the metabolic machinery in health and diseases.

Apart from the comparisons between disease and control, it will be interesting to study metabolic changes associated with dietary intervention in AMD [251,252]. A recent study on an animal model investigated this [130]. Plasma and urine samples were obtained from wild-type mice, which were fed diets with either a high- or low-glycemic index and subsequently analyzed by LC-MS and proton NMR. A total of 330 metabolites were found in plasma and urine (309 in plasma, 47 in urine, with 26 found in both). The mice fed the high-glycemic diet showed higher levels of lipids, including phosphatidylcholine, C3 carnitine, and lysoPE. Higher levels of the protein derivatives of 2-ω-carboxyethyl pyrrole (CEP) were also found [253,254]. These changes were associated with dysfunction of the RPE and degradation of the retinal cell structures. Therefore, dietary studies relevant for AMD can now be investigated both in animals and perhaps in humans [255].

Identifying biomarkers for such a multifactorial disease as AMD remains a significant challenge. There are noticeable overlaps between risk factors for AMD and many other diseases (as reviewed by Kersten et al. [17]). Therefore, metabolomics investigations can provide a further source of information that can be integrated into distinguishing AMD from other comorbidities. Further studies will shed light on potential metabolites, which may be used as early stage biomarkers, perhaps even being applied as precision medicine tools in the future treatment of AMD [256,257,258]. 

## Figures and Tables

**Figure 1 metabolites-09-00004-f001:**
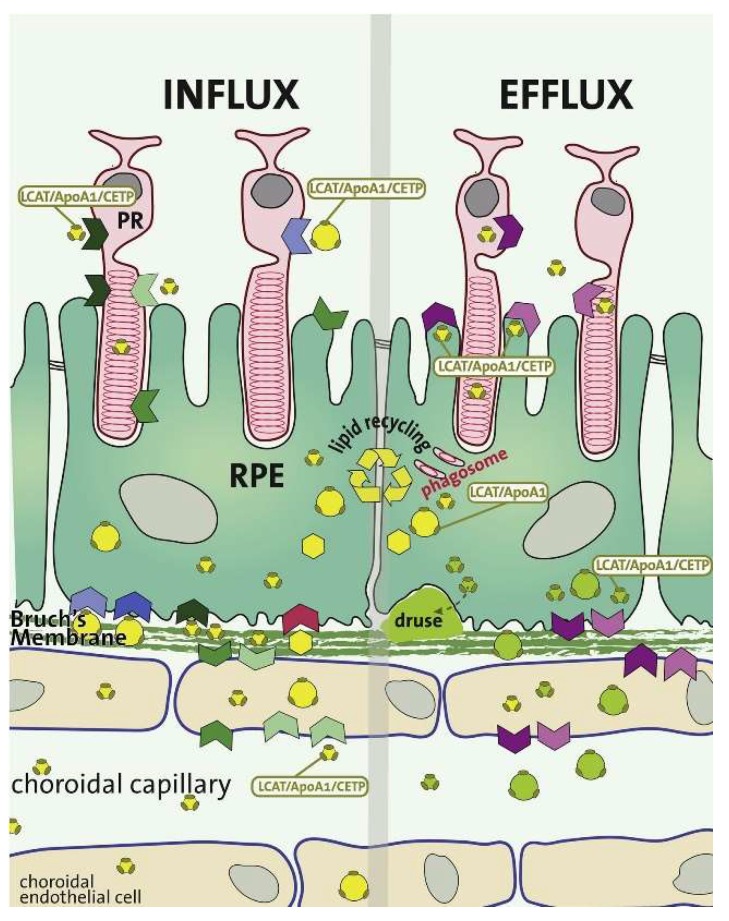
The metabolic flux of various lipids and their associated proteins in the retinal pigment epithelium (RPE). In addition to anabolic and catabolic lipid metabolism, the RPE also functions as a transfer site for lipids and proteins (yellow and green spheres) between the circulation and the photoreceptors. The influx of lipids from the RPE to the photoreceptors is represented on the left side, whilst the efflux of lipids is represented on the right side of the image. There is substantial recycling of lipids by the RPE, which are continuously provided through the phagocytosed membrane discs of photoreceptor outer segments (POS). The oxidized lipid species either enter the circulation as lipoprotein particles (green spheres) or are basally deposited into the sub-RPE space leading to the formation of drusen. LCAT, lecithin-cholesterol acyltransferase; APOA1, apolipoprotein A1; CETP, cholesteryl ester transfer protein. Colored arrows represent lipid receptors and their direction of transport. Reproduced with permission from van Leeuwen et al. [18].

**Figure 2 metabolites-09-00004-f002:**
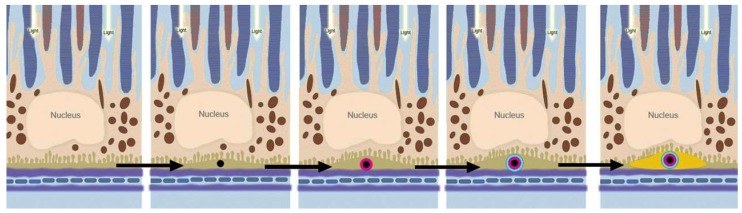
Model of sub-RPE deposit formation. Graphical overview of the proposed mechanism for the growth of sub-RPE deposits containing hydroxyapatite (reproduced with permission from Thompson et al. [89]). Micrometer-sized, cholesterol-containing extracellular lipid droplets (black) provide a site of hydroxyapatite (HAP) (magenta) precipitation. Deposit growth follows, with the binding of various proteins (blue) to the surface of HAP, facilitating a self-driven oligomerization process forming the macroscopic sub-RPE deposits (yellow). The brown particles within the RPE represent melanocytes.

**Figure 3 metabolites-09-00004-f003:**
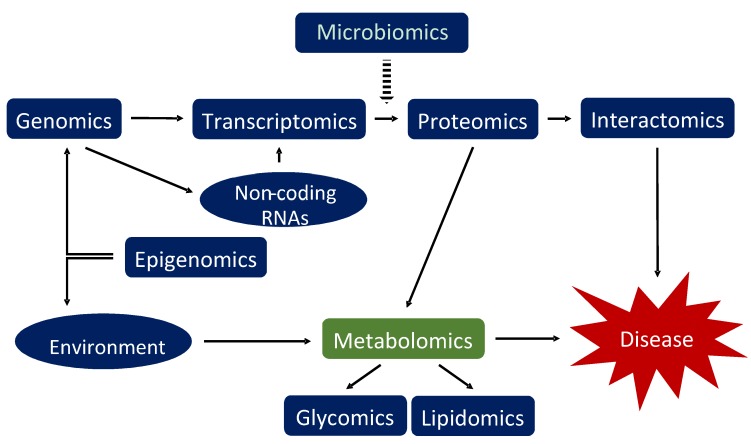
Overview of the various omics approaches that can be applied to assess the biological components that contribute to multifactorial diseases, such as age-related macular degeneration (AMD). Metabolomics is highlighted in green as an approach to test the effect of the environment and processes in the body on the development of disease. Microbiomics is growing in importance in understanding disease, but how it influences the whole system in a human, or in fact in animals, is yet to be determined. Adapted from Lauwen et al. [110].

**Table 1 metabolites-09-00004-t001:** Summary of untargeted metabolomic studies investigating AMD, including details of the tissue/biofluid profiled, and methods and instrumentation used.

Subjects and Biofluid Used	Number of Identified Metabolites	Metabolite Separation Method (Chromatography)	Detection Instrument Used	Reference
Mouse eye lysates	Not reported	Imtakt Scherzo SM-18 150 × 2 mm column Agilent 1200 capillary LC	Agilent 6538 UHD-QTOF MS (ESI+/ESI-)	[36]
Mouse eye lysates	203	XBridge C18 column (3.5 µm, 135 Å, 150 mm × 1.0 mm) Agilent 1260 HPLC	Agilent 6538 UHD Accurate Mass Q-TOF (ESI+)	[118]
Human serum (NVAMD); Mouse eye lysates	Not reported	Aeris Peptide XB-C18 column (3.6 µm, 100 × 2.10 mm) Phenomenex HPLC	Untargeted: Thermo Scientific LTQ Velos Orbitrap (ESI+/ESI-) Targeted: Thermo Scientific LXQ (ESI-)	[124]
hfRPE cells; Mouse retina; Apical/basal secretomes	202 (101 in medium; 53 changed substantially)	Ethylene bridged hybrid Amid column (1.7 µm, 2.1 mm × 150 mm) Agilent 1260 HPLC	AB Sciex QTrap 5500 (MRM)	[125]
Human plasma (NVAMD)	1168 (94 differed significantly)	Hamilton PRPX-110S (2.1 cm × 10 cm) anion exchange column	Thermo LTQ-FT spectrometer (ESI+)	[126]
Human plasma (NVAMD)	864 (10 differed significantly)	Acquity HSS T3 UPLC column (1.8 µm, 2.1 × 100 mm) Agilent 1290 Infinity UHPLC	AB SCIEX Triple 6600 TOF (ESI+/ESI-)	[127]
Human plasma	1188 spectra (30 low-M_w_ metabolites)	N/A	NMR: Bruker Avance DRX 500 spectrometer (300 K) operating at 500.13 MHz for protein, with a 5 mm TXI probe	[128]
Human plasma (nonexudative)	698 endogenous (87 associated with AMD)	C18 (acidic positive and basic negative ionization); HILIC (negative ionization) Waters ACQUITY ultra-UPLC (Metabolon, Inc.)	Thermo Scientific Q-Exactive (HESI-II), Orbitrap mass analyzer	[129]
WT mouse plasma (MS) and urine (NMR)	MS: 309 NMR: 47	Polar and nonpolar lipids: ACQUITY BEH C8 column (1.7 µm, 100 × 2.1 mm) Simadzu Nexera X2 UHPLC Hydrophilic metabolites: Atlantis HILIC column (3 µm, 150 × 2 mm) Shimadzu Nexera X2 UHPLC Additional polar metabolites: Phenomenex Luna NH2 column (150 × 2.0 mm) ACQUITY UHPLC	Polar and nonpolar lipids: Thermo Scientific Exactive Plus Orbitrap MS (ESI+) Hydrophilic metabolites: Thermo Fisher Scientific Q Exactive hybrid quadrupole Orbitrap MS (ESI+) Additional polar metabolites: AB SCIEX 5500 QTRAP MS (ESI- and MRM) NMR: Bruker Avance 600 spectrometer	[130]
Human plasma (NVAMD)	159 features differed (39 with medium to high confidence)	Hamilton PRP-X110S, 2.1 × 10 cm (anion exhchange) ^a^ Higgins Analytical C18 column, 2.1 × 10 cm (reverse phase) ^a^	Thermo Scientific LTQ Velos Orbitrap MS (ESI+)	[131]

^a^ Chromatography methods not stated, therefore, the above description was obtained from references provided in the study text.

**Table 2 metabolites-09-00004-t002:** Metabolites in human biofluids differing between AMD patients and control individuals. The number of individuals assessed in each study is indicated in parentheses.

Reference	Biofluid	Comparison	Metabolomic Technique Employed	Definitively Identified Metabolites	Level in AMD Cohort Compared to Controls
[124]	Blood serum	NVAMD patients (*n* = 22) and age-matched control patients (*n* = 22)	LC-MS	Docosahexaenoic acid	Lower
				Amino acids	Higher
				Prostaglandin G2	Higher ^NS^
[159]	Blood plasma	Intermediate AMD (drusen), late AMD (GA and CNV) (*n* = 77) and non-AMD control patients (*n* = 75)	HPLC and GC-negative-ion chemical ionization (NICI)-MS	Cystine Isofurans	Higher ^NS^
[126]	Blood plasma	NVAMD patients (*n* = 26) and age-matched control patients (*n* = 19)	LC- Fourier transform MS (FTMS)	Acetylphenylalanine	Higher
				Dipeptide; Tripeptides (modified cysteine and alanine ^a^)	
				Sethoxydim	
				Tripeptides ^b^	
				Tripeptides (acetyltryptophan^a^)	
				Flavones; halofenozide	
				Glycocholic acid	Lower
				Vitamin D-related metabolites; phytochemicals ^b^	
				Glycodeoxycholic acid+H^+^; Glycoursodeoxycholic acid+H^+^	
				Glycodeoxycholic acid+Na^+^; Glycoursodeoxycholic acid+Na^+^	
				Sencrassidol	
				Didemethylsimmondsin	
				Dipeptides ^b^	
[127]	Blood plasma	NVAMD patients (*n* = 20) and age-matched control patients (*n* = 20)	UPLC-TOF-MS	*N*-Acetyl-l-alanine	Higher
				l-Tyrosine	
				l-Phenylalanine	
				l-Methionine	
				l-Arginine	
				Isomaltose	
				Ν1-Methyl-2-pyridone-5-carboxamide	Lower
				l-Palmitoylcarnitine	
				Hydrocortisone	
				Biliverdin	
[167]	Blood serum	NVAMD patients (*n* = 20), PCV patients (*n* = 20), and age-matched controls (*n* = 20)	UPLC-QTOF-MS	Glycerophospholipids ^c^	Higher
				Phosphatidylcholine	
				Covalently modified amino acids ^c^	
				Di/tri-peptides ^c^	
				Tripeptides ^c^	
				ω-3 and ω-6 PUFAs ^c^	
				Pinolenic acid	
				Docoxahexaenoic acid	
				Eicosatetraenoic acid	
				Carnitine sp. ^c^	
[129]	Blood plasma	AMD patients (*n* = 314) and age-matched controls (*n* = 82), both across two locations	HILIC- and UPLC-MS	Creatine ^d^	Higher
				Oleic acid ^d^	Higher ^f^
				N(CH_3_)_3_ choline HDL ^d^	
				Acetate ^d^	
				Dimethylsulfone ^d^	
				Pyruvate ^d^	Higher ^g^
				Glutamine ^e^	
				Unsaturated F.A. ^e^	
				Unsaturated F.A. LDL + VLDL ^e^	
				Unsaturated F.A. ^d^	Lower ^f^
				Unsaturated F.A. LDL + VLDL ^d^	
				Histidine ^d^	Lower ^h^
				Acetoacetate ^d^	
				β-hydroxybutyrate ^d^	
				Unsaturated F.A. LDL + VLDL ^e^	Lower ^f^
				Glutamine ^e^	Lower ^g^
				Histidine ^e^	
				CH_2_CH_2_COOR F.A. ^e^	
				CH_2_CH_2_C=C F.A. ^e^	
				Albumin lysil ^e^	
				Alanine ^e^	Lower ^h^
				Histidine ^e^	
				Glyceryl C1,3H’ ^e^	
[128]	Blood plasma	AMD patients (*n* = 89) and age-matched control patients (*n* = 30)	NMR	*N*2-methylguanosine	Higher
				1-Stearoyl-2-oleoyl GPC	Lower ^NS^
				1-Linoleoyl-2-arachidonoyl GPC	Lower
				Stearoyl-arachidonoyl glycerol	
				Oleoyl-olyeol-glycerol	
				Dihomo-linolenoyl carnitine	
				1-Stearoyl-2-arachidonoyl GPC	
				Linoleoyl-linolenoyl glycerol	
				1-Stearoyl-2-linoleoyl-GPI ^f^	
				Oleoyl-linoleoyl-glycerol	
				Oleoylcarnitine	
				Ximenoylcarnitine	
				1-Stearoyl-2-arachidonoyl GPI ^i^	
[131]	Blood plasma	NVAMD patients (*n* = 100) and control patients (*n* = 192)	LC-MS and LC-MS/MS	l-Oxalylalbizziine ^j^	Higher
				Isopentyl beta-d-glucoside ^j^	
				LysoPC(P-18:0) ^j^	
				LysoPC(P-18:1(9Z)) ^j^	
				LysoPC(16:1(9Z)) ^j^	
				Darunavir ^j^	
				Bepridil ^j^	
				912-Hexadecadienoylcarnitine ^j^	
				456-Trimethylscutellarein 7-glucoside ^j^	
				1-Lyso-2-arachidonoyl-phosphidate ^j^	
				Americanin B ^j^	
				Corchoroside A ^j^	
				*N*-Ornithyl-l-taurine ^j^	
				Lyciumoside III ^j^	Lower
				Phosphatidylethanolamine ^f,j^	
				Phytosphingosine ^j^	
				Lenticin ^j^	
				9-Hexadecenoylcarnitine ^k^	Higher
				Heptadecanoyl carnitine ^k^	
				11Z-Octadecenylcarnitine ^k^	
				l-Palmitoylcarnitine ^k^	
				Stearoylcarnitine ^k^	
[168,169]	Blood serum and urine	Neovascular and nonexudative AMD patients (*n* = 104) ^l^	NMR	Arginine	Higher ^m,NS^
				Glucose	Lower ^m,NS^
				Lactate	
				Glutamine	
				Reduced glutathione	

^a^ When no specific metabolites were given (e.g., dipeptides), correlated Metlin matches with the same *m/z* are given; ^b^ No correlated Metlin matches; ^c^ Full list of 197 differing metabolites can be found in Supplementary Table 1 of [167]; ^d^ Metabolites obtained from the Coimbra cohort studied; ^e^ Metabolites obtained from the Boston cohort studied; ^f^ Effect size difference between early AMD patients vs controls; ^g^ Effect size difference between intermediate and early AMD patients; ^h^ Effect size difference between late and intermediate AMD patients; ^i^ No metabolite identity given; ^j^ Metabolites identified through high-resolution LC-MS; ^k^ Metabolite identity confirmed through high-resolution LC-MS/MS; ^l^ Specific patient cohort information not available; ^m^ When comparing NVAMD patients to nonexudative AMD patients. Abbreviations: AMD = age-related macular degeneration; F.A. = fatty acids; GA = geographic atrophy; GPC = glycerol-3-phosphocholine; GPI = glycosylphosphatidylinositol (assumed, no definition given); HDL = high-density lipoproteins; LC-MS = liquid chromatography-mass spectrometry; LDL = low-density lipoproteins; NVAMD = neovascular AMD; VLDL = very low-density lipoproteins; NS = not significant.

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
