# Peer review of "Metabolomics and Age-Related Macular Degeneration"

_metabolites, 2018, doi:10.3390/metabo9010004_

Reviewer 1 Report

This manuscript is a comprehensive review of metabolomics in age-related macular degeneration. The review is detailed, inclusive, and will serve as a good reference for the field. Some minor comments:

1. Line 38: I would add “advanced” prior to “AMD”….the other form of advanced AMD is geographic atrophy

2. Line 134: “A2E is preferentially located in the peripheral, rather than the central area, of human RPE.” It is unclear what “peripheral” and “central” are referring to. The retina? The macula? The macula vs. the peripheral retina? I realize that this is in the title of the reference, but it is not clear there either. This can be made more clear to the reader.

3. Line 147: “has” should be “have” for subject-verb agreement

4. Lines 169-170: “for several decades. In this study….” That doesn’t make sense. The authors state AGEs have been implicated in AMD progression for several decades and then say “in this study….” Perhaps “In one study in….”

5. Line 185: I would recommend changing “only” to “often.” AMD is often identified when patients have progressed beyond its initial stages, but not always.

6. Line 220: “was” should be “were” for subject-verb agreement

7. Line 221: add comma after “retina”

8. Lines 229-230: This is an awkward two sentence paragraph. The concept was mentioned earlier and is mentioned here again as if it is a new thought. Perhaps delete.

9. Line 257: “are” should be “is” for subject-verb agreement

10. Lines 533-538: The description of the patients is incorrect for this reference. They were all NVAMD patients (as the authors state in line 535). “exhibited extensive choroidal neovascularization, subretinal hemorrhaging, or geographic atrophy in one or both eyes” is not what is stated in the methods of the reference. There was no geographic atrophy, only NVAMD. Please check original reference.

11. Line 639: There is an extra space after “context” and there should be a comma after “context”

Author Response

Reviewer 1

This manuscript is a comprehensive review of metabolomics in age-related macular degeneration. The review is detailed, inclusive, and will serve as a good reference for the field. Some minor comments:

Thank you to Reviewer 1 for taking the time to read through our manuscript and for suggesting the changes below. They have been valuable in clarifying and improving the text throughout. Please find the implemented changes underneath each suggestion, which have also been incorporated into the manuscript through ‘Track Changes’.

1. Line 38: I would add “advanced” prior to “AMD”….the other form of advanced AMD is geographic atrophy

Line 38: The word “advanced” has been included before both cases of “AMD” in this sentence.

2. Line 134: “A2E is preferentially located in the peripheral, rather than the central area, of human RPE.” It is unclear what “peripheral” and “central” are referring to. The retina? The macula? The macula vs. the peripheral retina? I realize that this is in the title of the reference, but it is not clear there either. This can be made more clear to the reader.

Line 134-135: We agree that this required clarity. The text has been changed to indicate that “A2E is preferentially located in the RPE cells of the peripheral retina”, specifying the meaning of “peripheral” and “central” as highlighted in the reviewer’s comments. The rest of the sentence has been restructured accordingly.

3. Line 147: “has” should be “have” for subject-verb agreement

Line 147: “Has” was changed to “have”

4. Lines 169-170: “for several decades. In this study….” That doesn’t make sense. The authors state AGEs have been implicated in AMD progression for several decades and then say “in this study….” Perhaps “In one study in….”

Lines 169-171: We agree that the text can be viewed as contradictory. Reference 71 has been moved to the next sentence, as a direct link to that statement. The wording has also been changed from “…for several decades. In this study…” to “for several decades. In one early investigation…” to make the link between the two sentences clearer.

5. Line 185: I would recommend changing “only” to “often.” AMD is often identified when patients have progressed beyond its initial stages, but not always.

Line 185: “Only” was changed to “often”, thank you for identifying this oversight.

6. Line 220: “was” should be “were” for subject-verb agreement

Line 220: We could not find the identified “was” but the suggested change to “were” was already in the sentence.

7. Line 221: add comma after “retina”

Line 221: Comma added

8. Lines 229-230: This is an awkward two sentence paragraph. The concept was mentioned earlier and is mentioned here again as if it is a new thought. Perhaps delete.

Lines 214-216 and 230-232: Thank you for highlighting that we had previously mentioned zinc in Section 2.2. We have clarified this by restructuring the first paragraph of Section 2.3.4. to put zinc alongside calcium and a previously discussed metal, leaving iron alone as a newly included thought. The subsequent two sentence paragraph highlighted by the reviewer has been deleted (Lines 230-231) and the word “zinc” has been included in the subsequent paragraph to ensure the reader is aware of the change in metal ion discussion.

9. Line 257: “are” should be “is” for subject-verb agreement

Line 257: “Are” has been changed to “is”

10. Lines 533-538: The description of the patients is incorrect for this reference. They were all NVAMD patients (as the authors state in line 535). “exhibited extensive choroidal neovascularization, subretinal hemorrhaging, or geographic atrophy in one or both eyes” is not what is stated in the methods of the reference. There was no geographic atrophy, only NVAMD. Please check original reference.

Lines 536-538: Thank you for highlighting this mistake. Text has been added to include “subretinal haemorrhaging or fibrosis, or photocoagulation scarring”, whilst geographic atrophy has been removed.

11. Line 639: There is an extra space after “context” and there should be a comma after “context”

Line 639: The space has been deleted and the comma has been added

Reviewer 2 Report

This is an excellent and timely review! I enjoy reading this manuscript. Age-related macular degeneration (AMD) affects more and more people. It’s still one of the major causes for blindness.  The mechanism is poorly understood so far and the treatment is rather limited.  I also believe that metabolomics will play an important role in studying AMD. 

The authors started the review with a detailed description of metabolic events happening in the posterior part of the eye.   Then a comprehensive summary of the metabolomics studies of AMD was given including studies on retina tissues, RPE cells, biofluids, etc. In the last section, alternative approaches were suggested, for example, using tears, vitreous, aqueous humor and in vivo imaging.

I only spot one error in Table 2, ref. 164, the metabolomics technique used in ref. 164 is not NMR, but UPLC-MS/MS combining with RP and HILIC LC separation. I suggest the authors to check the references in the tables and make sure the integrity of the information.

Author Response

Reviewer 2

This is an excellent and timely review! I enjoy reading this manuscript. Age-related macular degeneration (AMD) affects more and more people. It’s still one of the major causes for blindness.  The mechanism is poorly understood so far and the treatment is rather limited.  I also believe that metabolomics will play an important role in studying AMD. 

The authors started the review with a detailed description of metabolic events happening in the posterior part of the eye.   Then a comprehensive summary of the metabolomics studies of AMD was given including studies on retina tissues, RPE cells, biofluids, etc. In the last section, alternative approaches were suggested, for example, using tears, vitreous, aqueous humor and in vivo imaging.

I only spot one error in Table 2, ref. 164, the metabolomics technique used in ref. 164 is not NMR, but UPLC-MS/MS combining with RP and HILIC LC separation. I suggest the authors to check the references in the tables and make sure the integrity of the information.

Thank you to Reviewer 2 for taking the time to read our manuscript, for the support of the manuscript and for suggesting the changes above. They have been valuable in ensuring the information in Table 2 was correct and, as suggested, the appropriate changes have been incorporated into the manuscript through ‘Track Changes’. These were for refs. 159, 126, 127, 165 and 166.

We chose not to include the extra details of RP and HILIC because these details had not been included for the other references as different detailed information was provided in Table 1.

Additional changes:

Line 366: Section numbering changed from “3.2.” to “3.3.”

Line 380: Changed from “a new insights” to “a new insight”

Line 407: Section numbering changed from “3.3.” to “3.4.”

Line 431: Section numbering changed from “3.4.” to “3.5.”